# Tracking Trace Elements Found in Coffee and Infusions of Commercially Available Coffee Products Marketed in Poland

**DOI:** 10.3390/foods13142212

**Published:** 2024-07-13

**Authors:** Kamila Pokorska-Niewiada, Aniela Scheffler, Laura Przedpełska, Agata Witczak

**Affiliations:** 1Department of Toxicology, Dairy Technology and Food Storage, Faculty of Food Sciences and Fisheries, West Pomeranian University of Technology, 71-459 Szczecin, Poland; kamila.pokorska@zut.edu.pl; 2Faculty of Food Sciences and Fisheries, West Pomeranian University of Technology, 71-459 Szczecin, Poland; anielasz1998@op.pl (A.S.); laura_przedpelska@o2.pl (L.P.)

**Keywords:** coffee, coffee infusions, trace elements, cadmium, lead, risk assessment

## Abstract

Coffee is a source of micronutrients, including iron, zinc, copper, and manganese. It may also contain toxic metals, such as lead and cadmium. The effects of coffee on the human body may vary depending on its composition. The objective of this study was to assess the quality of ground and instant coffee with regard to the content of selected trace elements. The concentrations of trace elements, including copper, iron, manganese, and zinc, were determined by ICP-AES, while the levels of lead and cadmium were quantified by GF-AAS methods. Furthermore, the degree of coverage of the recommended intake of elements and the risk assessment for human health (EDI, THQ, PTMI, and TWI) were determined. Our findings indicate that the consumption of a cup of coffee provides the body with only small amounts of these elements. A coffee prepared from 6.33 g of ground coffee beans provides 0.08–1.52% of the RDA value, while a coffee prepared from 6.33 g of instant coffee provides 0.46–13.01% of the RDA, depending on the microelement. The low transfer to the brew (Pb = 7.1%; Cd = 30.0%) of the analyzed ground coffees renders them safe for the consumer, even at a consumption of six cups per day. The percentage of benchmark dose lower confidence limit (BMDL_0.1_) in the case of lead did not exceed 0.9%. The estimated value did not exceed 0.2% of the provisional tolerable monthly intake of cadmium (PTMI). None of the analyzed coffees exhibited any risk regarding the trace elements.

## 1. Introduction

Coffee is one of the most widely consumed beverages worldwide and is appreciated for its flavor and aroma. Coffee beans are characterized by a rich chemical composition, with the main ingredient being carbohydrates (approx. 50–60%). In addition, they also contain, among others, proteins (approx. 13%), fats (approx. 15%), caffeine (approx. 1.5–2.3%), polyphenols (approx. 8%), minerals (approx. 7%), and other trace ingredients [1,2], which may affect human health.

Traditionally, coffee is shipped for distribution in the form of whole coffee beans or ground coffee [3,4]. In the case of instant coffee, the ground coffee undergoes extraction, evaporation, and drying, resulting in a dry, solid product that is then packaged and ready for distribution [5,6]. The beneficial effects of consuming small amounts of coffee, i.e., two to five cups per day, have been indicated in recent research. Such consumption has been found to help prevent many diseases of the cardiovascular, digestive, and nervous systems [7,8]. Drinking coffee can give a boost of energy and also improve mood. Coffee consumption has been shown to stimulate the movement that occurs in the gastrointestinal tract, and coffee drinking promotes the desire to have a bowel movement among around one-third of the population thanks to this stimulation of colonic motility [9].

This effect has been attributed to the presence of trace elements, which constitute 7% of coffee beans and which can play an important role in the normal functioning of the body. Iron, for example, is involved in oxygen transport, DNA synthesis, and electron transport. Zinc influences homeostasis, immune function, and oxidative stress [10]. Manganese is involved in the metabolism of proteins, fats, and carbohydrates and acts as a cofactor for enzymes such as decarboxylase, hydrolase, and kinase [11]. On the other hand, the presence of toxic elements can contribute to the development of health problems. Copper can be toxic depending on the amount consumed [12], lead causes cardiovascular, central nervous system, kidney, and fertility problems [13], and cadmium exerts multiple toxic effects by causing metabolic and structural changes in cells [14].

In Poland, approximately 60% of the adult population consumes coffee on a daily basis, with 16% of individuals drinking it at least twice a day. The most common form is ground coffee (45%), followed by instant coffee (40%) and then other alternatives such as bean coffee, coffee blends, or instant cappuccino [15]. Many people drink up to six cups of coffee per day. According to data from the Central Statistical Office, coffee consumption per capita in Poland in 2023 was 0.19 kg per month (6.33 g per day) [16]. For Poles, stimulating properties are very important, which is why they prefer to drink coffee with caffeine in the morning and at work. According to the Research and Development Centre [17], the Polish population is more likely to drink white coffee, i.e., with added milk or cream, than black coffee. White coffee consumption is highest among younger people, between the ages of 18 and 29 years of age, and this percentage decreases with age [17].

Epidemiological studies have associated higher coffee consumption with a reduced incidence of neurological and metabolic diseases, including Parkinson’s disease and type 2 diabetes. There is also evidence that coffee consumption reduces the incidence of colorectal, breast, and other cancers, although the results are conflicting [18,19]. Nevertheless, coffee consumption can also have harmful effects on health. Excessive, long-term caffeine consumption (more than 500–600 mg per day) can lead to addiction and various adverse symptoms, including excessive excitability, restlessness, arrhythmia, insomnia, headaches and stomach problems. Coffee can also have a negative effect on the health of people with mental illness, worsening symptoms or interfering with the effects of medication. It is also recommended that pregnant and breastfeeding women reduce their coffee intake and that for women who plan to become pregnant and during gestation, caffeine intake should not exceed 300 mg/day [20].

In addition, coffee beverages may be a source of contaminants; these can include toxic elements whose presence in the environment may be of natural (volcanic eruptions, leaching from soil and rocks) or anthropogenic origin (industrial and agricultural activities, including improper waste management) [21]. The mineral content of coffee, which can include toxic elements such as cadmium (Cd) and lead (Pb), can vary depending on the variety and type of coffee or the brewing method [22,23].

The aim of this article was to determine the content of certain trace elements, viz. copper, iron, zinc, manganese, lead, and cadmium, in coffees available on the Polish market and provide a comparison of the intake with the RDA (Recommended Dietary Allowance) of elements analyzed in ground coffee and instant coffee. It also estimates the health risk to consumers associated with exposure to these elements through drinking coffee.

## 2. Materials and Methods

The study analyzed instant and ground coffees from eight coffee producers; all were commonly available on the Polish market. The origin of the coffees and their blends is summarized in Table 1. Each coffee assortment came from five product batches, with different serial numbers (Figure 1). For analysis, 0.500 ± 0.001 g of coffee (as received) was taken and subjected to microwave-assisted mineralization with 5 mL (instant coffee) and 5 mL (ground coffee beans) of nitric acid (69% HNO_3_, Merck KGaA, Darmstadt, Germany). Coffee infusions were prepared by weighing 6.330 ± 0.001 g of ground coffee beans (as received) as well as 6.330 ± 0.001 g of instant coffee (as received) and pouring 100 mL of hot deionized water over them. Deionized water was prepared using an Easy Pure UV instrument (0.05 μS/cm; Barnstead ™ GenPure ™ Pro, Thermo Scientific, Dubuque, IA, USA). Coffee brewing time was 10 min. After cooling the infusions to 20 °C, 10 mL was taken for mineralization with 5 mL of nitric acid (69% HNO_3_, Merck KGaA, Darmstadt, Germany).

In order to ensure high quality analyses, in each mineralized series, a coffee sample, certified in accordance with reference material INCT-TL-1 (Institute of Nuclear Chemistry and Technology-Tea Leaves; Polish certified reference material, Poland) (0.150 ± 0.001 g), a blank sample (5 mL 69% HNO_3_, Merck KGaA, Darmstadt, Germany), and a fortified sample (with the addition of a standard iron solution) were digested. Fortified samples were made due to the lack of Fe in the reference material used. In order to determine the recovery of Fe in ground and instant coffee, an additional 12 samples of instant coffee and 12 samples of ground coffee were prepared (0.500 ± 0.001 g), including brands (A–H). A total of 0.1 mL of Fe standard solution (Merck KGaA, Darmstadt, Germany) with a concentration of 0.1 g/L was added to each powder sample (100% recovery corresponds to a concentration of 10 µg/sample). The sample was mixed thoroughly. In order to determine the recovery of Fe in coffee infusions, an additional 12 samples of instant coffee and 12 samples of ground coffee were prepared (coffee infusions prepared from 6.330 ± 0.001 g), including brands (A-H), which were fortified by adding 1 mL of Fe standard solution (Merck KGaA, Darmstadt, Germany) to each sample with a concentration of 0.01 g/L (100% recovery corresponds to a concentration of 100 µg/100 mL). Samples certified in accordance with the reference material and fortified samples were mineralized in parallel using the same procedure.

After digestion, the vessels were allowed to cool to room temperature. The contents were filtered if necessary (using a Whatman No. 40 filter; Merck KGaA, Darmstadt, Germany).

The statistical yearbook [16] only includes information on the amount of coffee consumed by Poles per day. There is no separation into types of coffee, whether it is ground or instant. To standardize the description of the results, the amount of coffee consumed was taken into account (6.330 g) and added with 100 mL of water.

Microwave mineralization was performed in an MDS-2000 oven (CEM Corp., Matthews, NC, USA) using the following parameters: maximum power of 1000 W, maximum pressure of 13.8 bar, maximum operating temperature of 200 °C, and total mineralization time of 90 min. Following the completion of the mineralization process, the samples were cooled, filtered, and quantitatively transferred to a 25 mL volumetric flask with deionized water.

Zinc, copper, manganese, and iron contents were determined by inductively coupled plasma atomic emission spectrometry (ICP-AES, Jobin Yvon JY-24, Jobin Yvon, Barnstable, MA, USA) using the following operating parameters: output power 1000 W, frequency 40.68 MHz, and plasma gas, auxiliary gas, and nebulizer gas composed of argon, with flow rates of 12.0, 1.0, and 1.1 mL/min, respectively. Cadmium and lead contents were determined by flameless atomic spectrometry (GF-AAS, Perkin-Elmer ZL 4110, Bodenseewerk Perkin-Elmer GmbH, Wellesley, MA, USA) with electrothermal atomization in a graphite cuvette. Five distinct furnace temperature stages were employed, spanning a range from 110 °C to 2500 °C. Atomization was conducted at 1600 °C for Pb and 1550 °C for Cd. The analysis used a slit width of 0.7 mm and a lamp current of 10 mA for Pb and 4 mA for Cd. The following wavelengths (nm) were applied: Zn—213.9; Cu—237.4; Fe—238.2; Mn—257.6; Cd—228.8; and Pb—283.3.

The recovery values of the determined elements were as follows: Zn—97.9 ± 1.1%; Cu—98.0 ± 0.9%; Fe—96.6 ± 1.7%; Mn—98.3 ± 1.2%; Cd—98.1 ± 0.9%; and Pb—99.1 ± 0.8%. The LOD (limit of detection) and LOQ (limit of quantification) of the analyzed trace elements were determined from the standard deviations (SDs) of the blank samples: LOD = 3 × SD, and LOQ = 10 × SD. The LOD and LOQ of the elements were (μg/kg) Zn (3.1; 10.3), Fe (1.2; 4.1), Mn (0.8; 2.6), Cu (2.9; 9.7), Cd (0.11; 0.37), and Pb (1.3; 4.4).

To determine the proportion of the RDA of zinc, iron, copper, and manganese provided by the consumption of coffee infusions, the infusions were prepared in accordance with the previously provided information, and the results were compared with the current dietary standards for the Polish population [24].

The human health risk assessment of the intake of elements with coffee was based on the Estimated Daily Intake (EDI) (1) and the Target Hazard Quotient (THQ) (2, 3). For assessment of toxic element intake (Pb, Cd) the standard values proposed by EFSA were adopted: TWI (tolerable weekly intake) for Cd (4)—2.5 μg/kg bw per week. Due to the damage caused by Cd to human health, the Joint FAO/WHO Expert Committee on Food Additives (JECFA) established a provisional tolerable monthly intake (PTMI) (5) of 25 μg/kg BW and a BMDL value (benchmark dose lower confidence limit) for Pb (6), determined for adult consumers: BMDL_01_—1.5 μg/kg/bw per day [25].

Estimated Daily Intake (EDI) [26]

EDI = (MS × C)/BW [µg/kg bw/day](1)
where MS is the daily coffee intake—6.33 g/day [16] with 100 mL of water; C is the content of the element in the analyzed coffee (mg/kg); BW is the reference body weight (70 kg).

b.Estimated Weekly Intake (EWI) [26]

EWI = EDI × 7 [µg/kg bw/week]

c.Target Hazard Quotient (THQ) [27]

THQ = (EF × ED × MS × C)/(RfD × MS × AT) × 10^−3^(2)
where EF is the exposure frequency to trace elements (365 days/year); ED is the exposure duration (70 years); MS is the daily coffee intake—6.33 g/day [16]; C is the concentration of trace elements in coffee (mg/kg); RfD is the oral reference dose of trace element (mg/kg bw/day): Zn = 0.3; Cu = 0.04; Fe = 0.7; Mn = 0.14; Cd = 0.001 [28]; BW is the reference body weight (70 kg); AT is the averaged exposure time to non-carcinogenic trace elements (365 days × 70 years).

d.Total Target Hazard Quotient (TTHQ)—the sum of the THQ of all elements analyzed

TTHQ = THQ (Fe) + THQ(Zn) + THQ(Cu) + THQ(Mn) + THQ(Cd)(3)

e.Tolerable weekly intake (TWI)

%TWI = (EWI_Cd_ × 100)/TWI(4)
where the TWI value for Cd = 2.5 µg/kg bw per week [29].

f.Provisional Tolerable Monthly Intake (PTMI)

%PTMI = (EDI_Cd_ × 30 × 100)/PTMI(5)
where the PTMI value for cadmium = 25 µg/kg bw per month [30].

g.Benchmark Dose Lower Confidence Limit (BMDL)

%BMDL_0.1_ = (EWI_Pb_ × 100)/BMDL_0.1_(6)
where the BMDL_0.1_ for Pb = for effects on systolic blood pressure in adults = 1.50 μg/kg bw per day [31].

The statistical analysis was conducted using StatSoft Statistica 13.0 (StatSoft Polska Sp. z o.o., Kraków, Poland) and Microsoft Excel 2017 (Microsoft, Poland). Correlations between variables were determined using Pearson’s correlation coefficient. One-way analysis of variance (ANOVA) and Tukey’s post hoc test were employed to assess the differences between the studied parameters. All differences were considered significant at *p* ≤ 0.05.

## 3. Results

The mean values and the ranges of the tested elements are given in Table 2a,c. Taking into account the trace elements Fe, Zn, Cu, Mn, the mean contents in ground coffee ranged from 6.0 ± 0.6 µg/g (Zn) to 38.1 ± 4.7 µg/g (Fe) (Table 2a). Copper is the only element analyzed whose concentration in instant coffee is much lower than in ground coffee beans.

The contents of the toxic elements lead and cadmium in coffee were significantly lower compared to trace elements, at 0.014–0.015 µg/g (ground coffee) and 0.015–0.017 µg/g (instant coffee), respectively (Table 2a).

Taking into account the percentage of transfer to infusions (Table 2b), the content of trace elements in drinks prepared from 6.33 g of ground coffee ranged from 3.5 ± 0.5 µg/100 mL (Cu) to 24.5 ± 1.9 µg/100 mL (Mn) (Table 2c).

Despite the 30% transfer of Cd to the infusion obtained from ground coffee (Table 2b), the contents found in the infusion were low, on average, 0.031 ± 0.003 µg/100 mL (Table 2c).

The RDA for men is higher for Zn and Mn, while for iron the RDA is higher for women. For copper, the RDA is the same for men and women. It was noticed that instant coffee, compared to ground coffee beans, covers the %RDA of the tested trace elements to a significantly greater extent (*p* < 0.05) (Table 3).

The mean Estimated Daily Intake (EDI) of each element in a cup of coffee (6.33 g coffee + 100 mL water) is considerably higher (except copper) for instant coffees (Table 4), as higher levels leach into the brew (100%) during the brewing process compared to ground coffee (Table 2b).

The lowest EDIs were found for lead and cadmium, and the highest for iron and manganese. The consumption of one cup of either ground or instant coffee per day poses a low risk of intake of toxic elements (Table 5 and Table 6).

The THQ factor describes the non-carcinogenic health risk of exposure to a toxic element; a THQ value greater than 1 indicates a chance that non-carcinogenic effects may occur, with a probability that tends to increase with the THQ value [32]. However, the Target Hazard Quotients (THQs) were significantly lower than one for both ground and instant coffee (Table 5). It was concluded that there is no risk associated with non-carcinogenic exposure to these elements.

Despite the TTHQ coefficient in instant coffee being over seven times higher than in ground coffee beans (Table 5), it does not pose a risk to the consumer (in terms of trace elements), as even the maximum TTHQ value in instant coffee is well below one.

## 4. Discussion

Due to its widespread popularity, it is of great importance to be aware of the chemical composition of coffee. Also, while coffee is not typically presented as a source of minerals, given its frequent consumption, it can also serve as a supplement for certain minerals [33].

The average iron content in ground coffee found in this study was similar to the results obtained by other authors, including Grembecka et al. [34] (41.6 µg/g) and Adler et al. [35] (41.1 µg/g), although higher levels were reported by Ashu and Chandravanshi [36], with a mean content of 53 µg/g. In the case of instant coffee in this study, the average iron content was higher than the values obtained by Grembecka et al. [34], who reported a mean content of 34 µg/g, and lower than those noted in previous studies, which range from 51.2 to 84.7 µg/g [37,38,39]. The highest concentration, exceeding 450 µg/g, was reported by Dos Santos and Oliveira [40]. Polish standards [25] recommend an iron intake of 18 mg per day for women and 10 mg per day for men; therefore, one cup of ground coffee (containing 6.33 g of coffee), brewed by pouring boiling water, provides only about 0.10% of the RDA for women and 0.18% for men, and one cup of instant coffee (containing 6.33 g of coffee) provides almost 3% of the daily iron requirement for men. The consumption of ground coffee infusion resulted in a target hazard quotient (THQ) of no higher than 0.0004, which was significantly lower than the threshold value of 1, while instant coffee had a value of no more than 0.0061. Unfortunately, there are no comparable references to these data in the available literature.

The mean zinc content of the tested ground coffees was 6.0 ± 0.6 µg/g. Similar results were obtained by Adler et al. [35]. The zinc content of ground coffee can vary considerably, with values ranging from 1.2 µg/g to as much as 803 µg/g [41]. The mean zinc content of the tested instant coffees was 9.18 ± 0.75 µg/g; this is roughly twice that noted by Grembecka et al. [34]. Other authors have recorded higher contents, with values ranging from 16–18 µg/g [39] to 26.5–37.7 µg/g [38]. Polish standards [25] recommend a zinc intake of 8 mg/day for women and 11 mg/day for men. One cup of ground coffee fulfills only at most 0.15% of this requirement, and in the case of instant coffee, it is no more than 0.82%. The target hazard quotient (THQ) values of zinc were lower than those reported by Anissa et al. [42], with a THQ value of 0.012 for ground coffee.

In our studies, the copper content in ground coffee was much higher compared to instant coffee. Moreover, the mean copper content of the tested ground coffees were higher than the those obtained by Gogoasa et al. [39], which range from 8.2 to 9.9 µg/g, but lower than those of Anthemidis and Pliatsika [43], who note a maximum value of 12.2 µg/g. Additionally, Cruz et al. [44] reported values ranging from 9 to 31 µg/g.

In the present study, the concentrations of copper in instant coffees were significantly higher than those reported by Grembecka et al. [34], who found a mean value of 0.07 µg/g, while other authors [40] report values ranging from 0.5 to 2.33 µg/g. The highest concentration of copper was observed by Cruz et al. [44], reaching up to 16 µg/g. Polish standards [25] indicate the RDA for copper to be 0.9 mg. Hence, one cup of ground coffee infusion will cover not more than 0.49% of the RDA for both men and women, while a cup of instant coffee will cover 1.04%. The maximum THQ value for copper was 0.0015 for ground coffee and 0.0033 for instant coffee. While Anissa et al. [42] report a THQ level for ground coffee of 0.0022, no comparable values currently exist for instant coffee.

The average concentration of manganese in the analyzed ground coffees was comparable to the findings of other authors [34,36,43,45]. However, significantly higher manganese concentrations in coffee were recorded by Zaidi et al. [38], with values ranging from 24.6 to 49.5 µg/g, i.e., approximately three times higher than the result obtained in the present study. In the present study, the mean manganese content of the tested instant coffee was 28.4 ± 5.4 µg/g. In contrast, a range of values have been reported by other authors, ranging from 3.62 mg/kg to 38.85 µg/g [34,37,38,40]. In accordance with the current Polish standards [25], the daily manganese requirement is 1.8 mg for women and 2.3 mg for men. Hence. one cup of ground coffee infusion covers maximally 1.52% of the RDA for women and 1.19% for men, while a cup of instant coffee provides 13.3% of the RDA of manganese for women and 10.2% for men. The maximum manganese THQ coefficient was 0.0028 for ground coffee and 0.0239 for instant coffee; these values are almost three times lower than those obtained by Anissa et al. [42].

The mean lead concentration in ground coffees was comparable with results obtained by other authors [35,39], who found a mean lead concentration of 0.017 µg/g and 0.02 µg/g, respectively. Low levels of lead were also found in instant coffee, averaging 0.014 ± 0.002 µg/g. The concentration of lead in instant coffee was significantly lower than that obtained by Pohl et al. [41], who found the mean value to range from 0.09 to 0.91 µg/g.

The mean cadmium content of ground coffees was found to be low, at 0.017 ± 0.002 µg/g. Similar results were obtained by Adler et al. [35], with a mean cadmium content of 0.014 µg/g. In contrast, Gogoasa et al. [39] found the content to be much lower, at <0.002 µg/g, and Santos et al. [45] found it to be below 0.1 µg/g. Regarding instant coffees, the cadmium content was higher than the concentration recorded by Winiarska-Mieczan et al. [46] and lower than other authors [34,41]. The THQ values for cadmium in the present study were low and indicate a negligible risk to consumer health, assuming the consumption of one cup of coffee per day (prepared from 6.33 g of ground coffee beans or instant coffee). Slightly higher results were obtained by Kowalska [47] and Winiarska-Mieczan et al. [46] for ground coffee, while Anissa et al. [42] obtained a considerably higher score of approximately 2.2.

It can be seen that the profiles of the coffees examined differ from those reported by other authors, and this may be attributed to a number of factors. Primarily, the content of trace elements in coffee beans is influenced by the geographical region, the soil type, the growing conditions [48], and also by the technological processes used in coffee production, such as drying, roasting, or storage. Dippong et al. [49] report significantly lower mineral content in an Arabica variety than in a Robusta variety, with the mineral (K, Ca, Mg, Fe, Cu, P, N, and S) content increasing with roasting intensity. It is important to note that any toxic elements [35], inorganic fertilizers, and organic residues taken up by the plants from the soil [41] can be transferred to the beverage during the brewing process, together with the mineral components [35].

Instant coffee undergoes a slightly different production process to ground coffee, insofar that roasting, grinding, and extraction are followed by concentration to a dry powder. Consequently, the quantity of elements present in the final product is primarily determined by the initial content of the coffee beans. It is important to note that while some minerals are naturally more water soluble, others form complexes with the matrix, which reduces their extractability [44]. Furthermore, the content of minerals in the final coffee brew is influenced by similar factors to those affecting the composition of coffee beans but also by the method of brewing, the type of water used, the brewing time, the ratio of coffee to water, and the degree of roasting and grinding of coffee beans; it can also be determined by pressure in the case of coffee machine or coffee pot brews [22]. Our findings indicate that the daily intake of lead and cadmium from coffee brews is low, even assuming the consumption of three cups per day.

## 5. Conclusions

Coffee, although it contains microelements, is a poor source of them. Nevertheless, it has been proven that consuming one cup of coffee (6.33 g of coffee and 100 mL of water) provides the body with small amounts of manganese, iron, copper and zinc, differing with the method of preparing the coffee (ground coffee beans or instant coffee).

It has been demonstrated that coffee beans may contain harmful elements, including cadmium and lead. However, due to the relatively low transfer into the brew (Pb = 7.1%; Cd = 30.0%), the ground coffees analyzed were found to be safe for the consumer when three cups are consumed per day.

The strong point of this work is the comparison of the most popular coffees in Poland in terms of their elemental content, as well as determining the degree of risk associated with the intake of toxic elements. The weakness is the inability to take into account consumers’ preferences, as some people prefer stronger coffee infusions, and others weaker. Therefore, data included in the statistical yearbook were used. It is also worth expanding the research to include other coffees, as well as coffee additives, which are becoming more and more popular and may affect the mineral composition of coffee drinks.

## Figures and Tables

**Figure 1 foods-13-02212-f001:**
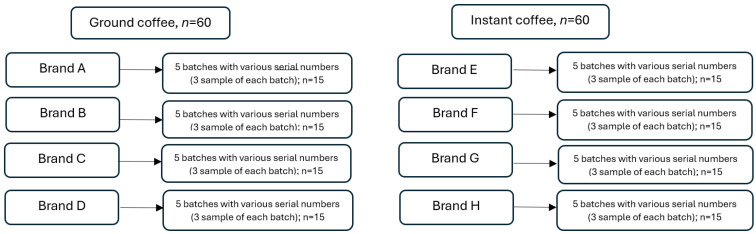
Sampling scheme for analysis.

**Table 1 foods-13-02212-t001:** Characteristics of the analyzed coffees.

Coffee Brands	Type of Coffee	Species
A	ground coffee beans	Arabica—50% Robusta—50%
B	ground coffee beans	Arabica and Robusta mix
C	ground coffee beans	Arabica—100%
D	ground coffee beans	Arabica and Robusta mix
E	instant coffee	Arabica and Robusta mix
F	instant coffee	Arabica and Robusta mix
G	instant coffee	Arabica and Robusta mix
H	instant coffee	Arabica and Robusta mix

**Table 2 foods-13-02212-t002:** (**a**) The concentrations of the tested elements present in coffee, divided into types [µg/g]. (**b**) The percent transfer of the tested elements to the brews from ground coffee [%]. (**c**) The concentrations of the tested elements present in coffee drinks, divided into types.

(a)							
Type of Coffee		Fe [µg/g]	Zn [µg/g]	Cu [µg/g]	Mn [µg/g]	Pb [µg/g]	Cd [µg/g]
ground coffee beans	mean ± SD range	38.1 ± 4.7 ^1^ 31.0–45.3 ^2^	6.0 ± 0.6 5.1–6.7	10.4 ± 1.6 8.5–12.6	15.8 ± 1.2 14.1–17.7	0.015 ± 0.004 0.009–0.020	0.017 ± 0.002 0.013–0.019
instant coffee	mean ± SD range	42.9 ± 2.6 39.1–46.8	9.2 ± 0.8 8.0–10.4	1.2 ± 0.1 1.0–1.5	28.4 ± 5.4 20.1–37.1	0.014 ± 0.003 0.010–0.020	0.015 ± 0.002 0.012–0.019
**(b)**							
**Type of Coffee Drink**		**% Transfer to Infusion**					
		**Fe**	**Zn**	**Cu**	**Mn**	**Pb**	**Cd**
ground coffee beans		7.23 ± 1.27 ^1^	28.12 ± 2.67	5.22 ± 1.63	25.08 ± 2.2	7.10 ± 1.63	30.05 ± 3.27
instant coffee		100	100	100	100	100	100
**(c)**							
**Type of Coffee Drink**		**Fe** **[µg/100 mL]**	**Zn** **[µg/100 mL]**	**Cu** **[µg/100 mL]**	**Mn** **[µg/100 mL]**	**Pb** **[µg/100 mL]**	**Cd** **[µg/100 mL]**
ground coffee drink	mean ± SD range	17.8 ± 2.2 ^1^ 14.5–21.2 ^2^	10.5 ± 1.1 8.9–11.8	3.5 ± 0.5 2.9–4.3	24.5 ± 1.9 21.9–27.4	0.007 ± 0.002 0.004–0.009	0.031 ± 0.003 0.025–0.035
instant coffee	mean ± SD range	271.4 ± 16.1 247.3–296.6	58.1 ± 4.7 50.7–65.6	7.8 ± 0.7 6.4–9.3	179.5 ± 34.0 127.5–234.6	0.090 ± 0.016 0.065–0.126	0.096 ± 0.014 0.078–0.118

^1^ Arithmetical mean ± SD (standard deviation) for each analyzed sample, *n* = 60 (according to Figure 1). ^2^ Minimum−maximum concentration.

**Table 3 foods-13-02212-t003:** Percentage of the Recommended Dietary Allowance (%RDA) of elements provided by consumption of coffee drinks prepared using 6.33 g coffee (based on Polish standards; [25]).

RDA-Polish Standards [25]		Fe		Zn		Cu	Mn	
		W * 18 mg/24 h	M ** 10 mg/24 h	W 8 mg/24 h	M 11 mg/24 h	W/M 0.9 mg/24 h	W 1.8 mg/24 h	M 2.3 mg/24 h
Type of Coffee Drink		%RDA						
ground coffee beans	mean ± SD range	0.10 ± 0.01 ^1^ <0.01–0.12 ^2^	0.18 ± 0.02 0.15–0.21	0.13 ± 0.01 0.11–0.15	0.10 ± 0.01 0.08–0.11	0.39 ± 0.06 0.32–0.48	1.36 ± 0.11 1.21–1.52	1.07 ± 0.08 0.95–1.19
instant coffee	mean ± SD range	1.51 ± 0.09 1.37–1.65	2.71 ± 0.16 2.47–2.97	0.73 ± 0.06 0.63–0.82	0.53 ± 0.04 0.46–0.60	0.87 ± 0.09 0.71–1.04	9.97 ± 1.89 7.08–13.03	7.80 ± 1.48 5.54–10.2

* Women; ** Men; ^1^ Arithmetical mean ± SD (standard deviation) for each analyzed sample, *n* = 60; ^2^ Minimum−maximum concentration.

**Table 4 foods-13-02212-t004:** Estimated Daily Intake (EDI) for coffee drinks made using 6.33 g of ground coffee/instant coffee.

Type of Coffee Drink		EDI [μg/kg bw/day]					
		Fe	Zn	Cu	Mn	Pb	Cd
ground coffee beans	mean ± SD range	0.25 ± 0.03 0.21–0.30	0.15 ± 0.02 0.13–0.17	0.05 ± 0.01 0.04–0.06	0.35 ± 0.03 0.31–0.39	0.0001 ± 0.0001 0.0001–0.0001	0.0004 ± 0.0001 0.0004–0.0005
instant coffee	mean ± SD range	3.88 ± 0.23 3.53–4.24	0.83 ± 0.07 0.72–0.94	0.11 ± 0.01 0.09–0.13	2.56 ± 0.49 1.82–3.35	0.0013 ± 0.0002 0.0009–0.0018	0.0014 ± 0.0002 0.0011–0.0017
	**EWI [μg/kg bw/week]**						
ground coffee beans	mean ± SD range	1.78 ± 0.22 0.21–0.30	1.05 ± 0.11 0.21–0.30	0.36 ± 0.03 0.21–0.30	2.45 ± 0.19 0.21–0.30	0.0007 ± 0.0002 0.0004–0.0009	0.0031 ± 0.0003 0.0025–0.0035
instant coffee	mean ± SD range	27.14 ± 1.62 24.73–29.66	5.81 ± 0.48 5.07–6.55	0.78 ± 0.08 0.64–0.93	17.95 ± 3.40 12.75–23.46	0.0090 ± 0.0016 0.0065–0.0126	0.0096 ± 0.0014 0.0078–0.0118

**Table 5 foods-13-02212-t005:** Estimated values for the Target Hazard Quotient (THQ) and Total Target Hazard Quotient (TTHQ).

Type of Coffee Dink	Range	THQ					TTHQ
		Fe	Zn	Cu	Mn	Cd	
ground coffee beans	min. max.	0.0003 0.0004	0.0004 0.0006	0.0010 0.0015	0.0022 0.0028	0.0004 0.0005	0.0049 0.0052
instant coffee	min. max.	0.0050 0.0061	0.0024 0.0031	0.0023 0.0033	0.0130 0.0239	0.0011 0.0017	0.0253 0.0373

**Table 6 foods-13-02212-t006:** Estimated percentage realization of %BMDL_0.1_, %TWI, and %PTMI with consumption of 1 and 3 cups of coffee per day.

Type of Coffee Drink		1 Cup of Coffee (6.33 g of Coffee)			3 Cups of Coffee (18.99 g of Coffee)		
		Pb	Cd		Pb	Cd	
		%BMDL_0.1_	%TWI	%PTMI	%BMDL_0.1_	%TWI	%PTMI
ground coffee beans	mean ± SD range	0.047 ± 0.012 0.029–0.062	0.124 ± 0.013 0.099–0.142	0.053 ± 0.005 0.042–0.061	0.140 ± 0.036 0.086–0.186	0.371 ± 0.038 0.296–0.426	0.159 ± 0.016 0.127–0.182
instant coffee	mean ± SD range	0.600 ± 0.107 0.430–0.840	0.385 ± 0.0056 0.314–0.471	0.165 ± 0.024 0.135–0.202	1.801 ± 0.321 1.291–2.519	1.156 ± 0.168 0.942–1.413	0.495 ± 0.072 0.404–0.606

## Data Availability

The original contributions presented in the study are included in the article, further inquiries can be directed to the corresponding author.

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
