# Peer review of "Tracking Trace Elements Found in Coffee and Infusions of Commercially Available Coffee Products Marketed in Poland"

_foods, 2024, doi:10.3390/foods13142212_

Round 1

Reviewer 1 Report

Comments and Suggestions for Authors

Foods-3082733: “Tracking trace elements found in coffee and infusions of commercially available coffee products marketed in Poland” by authors Kamila Pokorska-Niewiada, Aniela Scheffler, Laura Przedpelska and Agata Witczak.

The results of a multi-elemental analysis of trace elements in coffee from the Polish market are presented and discussed in the manuscript. They analysed 4 samples of ground coffee beans and 4 instant coffee powders. The coffee samples were a mixture of Arabica and Robusta variety, except for one pure Arabica sample. The authors determined the total concentration of Cu, Fe, Zn, Mn, Pb, and Cd in coffee samples after microwave digestion and their soluble fractions. Based on this data they estimated the consumer’s health risk associated with the exposure to potentially toxic elements during coffee consumption by calculating the Estimated Daily Intake, Target Hazard Quotient (THQ) and total THQ, and the Provisional Tolerable Weekly Intake. They showed that the concentrations of elements and consequently the intake parameters and hazard quotients were very low. All the coffee samples were found to be safe for consumption.

The manuscript is clearly written and well structured. Unfortunately, the number of samples and determined elements is very scarce. Additionally, the aim of the study is superficial. The selection of elements is questionable, why only Cu, Fe, Zn, Mn, Pb, and Cd? There are many reports about the multi-elemental composition of coffee with a much bigger elemental selection and number of samples available. 

Some specific remarks to be addressed:

-       please verify the percentage amount of mineral elements present in the coffee - the authors wrote that about 7 % of them are present but the cited references do not confirm this number;

-       why so small number of samples? Indeed, most of the coffees are a mixture of Arabica and Robusta varieties but they may be from different geographical origins, and therefore could account for a broader range of major and trace element concentrations; what about the difference between the grounded coffee beans and the coffee in beans, this could potentially point to the contamination during the preparation (milling) process, …

-       regarding sample preparation: was the same amount of ground beans and infusion powder used for MW digestion? Was 3 mL of nitric acid enough for total digestion of coffee beans?

-       please state the full name of the certified reference material used;

-       please add the full description of how the infusions were prepared, what kind of water was used, it was MilliQ or tap water. If tap water, what was the elemental composition of the water?

-       in Table 2 total concentrations are presented, please add the concentrations of the infusions as they are relevant for human consumption;

 Based on the above remarks, the manuscript in its present form is not suitable for publication.

Author Response

Dear Reviewer,

all authors would like to thank all the Reviewers and Editor for their comments and time spent on the manuscript evaluation. Thank you for giving us the opportunity to revise and improve our article. All the changes have been saved in blue in main manuscript. We hope that our answer will be sufficient to dispel the reviewer's doubts.

The text has been improved and checked by native speaker.

  1. Please verify the percentage amount of mineral elements present in the coffee - the authors wrote that about 7 % of them are present but the cited references do not confirm this number.

Thank you for pointing out - while editing the work, reference number 2 was incorrectly inserted. The manuscript has been corrected

  1. Why so small number of samples? Indeed, most of the coffees are a mixture of Arabica and Robusta varieties but they may be from different geographical origins, and therefore could account for a broader range of major and trace element concentrations; what about the difference between the grounded coffee beans and the coffee in beans, this could potentially point to the contamination during the preparation (milling) process, ..

When selecting samples for analysis, the preferences of Polish consumers were taken into account and the most popular coffee varieties and their producers were selected. Of course, the authors agree with the reviewer that it is worth extending the research to include whole grains as well as examining less popular varieties, but this will be the subject of further analysis.

  1. Regarding sample preparation: was the same amount of ground beans and infusion powder used for MW digestion? Was 3 mL of nitric acid enough for total digestion of coffee beans?

Thank you for pointing out - 3 ml of nitric acid was used to mineralize instant coffee, and 6 ml of nitric acid was used for ground coffee beans. The manuscript has been supplemented.

  1. Please state the full name of the certified reference material used.

The content of the manuscript has been supplemented

  1. Please add the full description of how the infusions were prepared, what kind of water was used, it was MilliQ or tap water. If tap water, what was the elemental composition of the water?

The following information has been added to the manuscript: Deionized water was used to prepare the infusions. It was prepared using an Easy Pure UV instrument (0.05 μS/cm; Barnstead ™ GenPure ™ Pro, Thermo Scientific, Dubuque,IA, USA).

  1. In Table 2 total concentrations are presented, please add the concentrations of the infusions as they are relevant for human consumption.

The content of the manuscript has been supplemented

Reviewer 2 Report

Comments and Suggestions for Authors

First some general comments. 1.  There are many instances where the word “amount” is used incorrectly for what is, in fact, a concentration.  “Amount” and “concentration” are definitely not synonyms: the former will have units such as mass, volume, number; the latter will have units such as mass fraction or mass per volume. 2.  Whenever a ± term is given, there should be an explanation of what this is.  If a standard deviation the number of replicates should be given, if a confidence interval the degree of confidence (such as 95%) and the number of replicates should be given. 3. It seems inappropriate to include a paragraph describing some of the adverse health outcomes of coffee consumption.  But if you leave this in, there needs to be a paragraph citing studies that show the benefits of coffee consumption. (see, for example, https://www.hopkinsmedicine.org/health/wellness-and-prevention/9-reasons-why-the-right-amount-of-coffee-is-good-for-you)

There are two major parts to this manuscript. 1. The determination of the concentrations of 6 elements in coffee (both ground and instant) in brands available in Poland, so that 2. The contribution of these elements to the dietary intake of Poles can be estimated.  My expertise is firmly in the area of the chemical analysis, but I will first raise some questions about part 2. 

The calculations all start with the estimated daily intake (EDI) (lines 128 and 129).  Surely for this calculation you need to know the concentrations of the elements in a cup of coffee, the volume of a cup, and how many cups are drunk per day?  And to know the relative proportions of drinks made from ground coffee and drinks made from instant coffee.  But the number that appears in the calculation is 6.33 g per day.  This is difficult to understand:  6.33 g of what?  And how does this relate to drinking coffee? 

In Table 3, the entries are just single numbers, so how were these calculated?  The results of the analyses of the coffees is given as a range of values (Table 2), then means were calculated? (data in Fig 1) and numbers in text (taking Fe as an example from line 209, which by the way, contains a example of an undefined ± term).  Then the fraction of the element extracted by 100 mL of hot water from 3 g of coffee needs to be taken into account?  This information is in lines 182 to 184.  But the term “concentration” is not appropriate, these numbers are the percentages of the mass of the element that are extracted.  It is hard to be sure, as the results of the analyses of these extracts are not given.  And by the way, the comment about Mn does not make sense, the percentage of Mn extracted (25) is relatively high.  So putting all this together, one does in fact come up with 0.08 as the percentage of the RDA (of 10 mg for men) delivered by the extraction of 3 g of ground coffee with 100 mL, and 1.28% of the RDA from the same amount of instant coffee (for which one has to assume that 100% of the Fe in the 3 g of instant coffee are consumed).  Would it not be more informative to express these %RDA numbers as a range? And you should state the assumption about the percentage extraction of elements from instant coffee (or give the actual percentages).

I cannot follow the calculation of the numbers in Table 4.  I’ve just looked at the Fe numbers.  One “3 g + 100 mL of water” cup of ground coffee delivers 38.1 x 3 x 0.072 µg of iron and so for a 70 kg adult the EDI is 0.12 µg per kg body weight.  The corresponding calculation for instant coffee gives 1.84 µg per kg body weight.  These do not agree with the numbers in the Table.  So some explanation of how the numbers in the Table were calculated is needed.

It might be a good idea to check the numbers in Table 5.

Chemical Analysis:  While it is of interest, no doubt, to know the concentrations of these elements in ground and instant coffees, in terms of the impact on the diet, it is the concentrations in the brewed coffee that is important.  First, though, a couple of comments about the analysis of the solids: did you account for the moisture content?  So the results are on a dry weight basis?  What about the reference material (RM)?  The certificate values will refer to dry weight.  This RM (tea leaves) is not a good match for theses coffees (and the iron value is not a certified value). Leaving aside the difference in matrix, the concentrations in the RM are many times those in your samples (e.g. the Mn content is 1570 mg/kg) and the analysis must have involved considerable dilution.  Some details should be provided.  Did you take 500 mg of the RM?

More importantly, details of the analysis of the extracts should be provided.  Were they filtered? Diluted? And what were the concentrations of elements found (in each of the 8 samples?).  How do you know these results are accurate and that there were no matrix effects?  The results of spike-recovery experiments should be provided.  Results should be presented to show that 100% of the elements are, in fact, extracted from instant coffee.

As a postscript, estimating the daily dietary intake based on one “cup” made from 3 g of coffee is surely a significant underestimation of the the reality.  Many individuals will surely consume the extract from a much higher mass?  As an example, I drink relatively weak coffee, but still consume the extract of about 15 g of ground coffee (a mixture of caffeinated and decaffeinated) per day (in a total volume of about 900 mL).

And finally, what about decaffeinated coffees?  Does the extraction of the caffeine impact the mineral content?

Comments on the Quality of English Language

For the most part, the English is fine.  The confusion between "amount" and "concentration" needs to be fixed and there are some errors.  For example, on line 251, the word "copper" should be manganese.

Author Response

Dear Reviewer,

all authors would like to thank all the Reviewers and Editor for their comments and time spent on the manuscript evaluation. Thank you for giving us the opportunity to revise and improve our article. All the changes have been saved in blue in main manuscript. We hope that our answer will be sufficient to dispel the reviewer's doubts.

The text has been improved and checked by native speaker.

First some general comments.

  1. There are many instances where the word “amount” is used incorrectly for what is, in fact, a concentration.  “Amount” and “concentration” are definitely not synonyms: the former will have units such as mass, volume, number; the latter will have units such as mass fraction or mass per volume.

The authors made corrections to the manuscript.

  1. Whenever a ± term is given, there should be an explanation of what this is.  If a standard deviation the number of replicates should be given, if a confidence interval the degree of confidence (such as 95%) and the number of replicates should be given.

The authors made corrections to the manuscript.

  1. It seems inappropriate to include a paragraph describing some of the adverse health outcomes of coffee consumption.  But if you leave this in, there needs to be a paragraph citing studies that show the benefits of coffee consumption. (see, for example, https://www.hopkinsmedicine.org/health/wellness-and-prevention/9-reasons-why-the-right-amount-of-coffee-is-good-for-you)

 The authors made corrections to the manuscript. From conclusions authors removed information about the properties of coffee, but added information in the introduction.

The calculations all start with the estimated daily intake (EDI) (lines 128 and 129).  Surely for this calculation you need to know the concentrations of the elements in a cup of coffee, the volume of a cup, and how many cups are drunk per day?  And to know the relative proportions of drinks made from ground coffee and drinks made from instant coffee.  But the number that appears in the calculation is 6.33 g per day.  This is difficult to understand:  6.33 g of what?  And how does this relate to drinking coffee? 

The authors made corrections to the manuscript.

In Table 3, the entries are just single numbers, so how were these calculated?  The results of the analyses of the coffees is given as a range of values (Table 2), then means were calculated? (data in Fig 1) and numbers in text (taking Fe as an example from line 209, which by the way, contains a example of an undefined ± term).  Then the fraction of the element extracted by 100 mL of hot water from 3 g of coffee needs to be taken into account?  This information is in lines 182 to 184.  But the term “concentration” is not appropriate, these numbers are the percentages of the mass of the element that are extracted.  It is hard to be sure, as the results of the analyses of these extracts are not given.  And by the way, the comment about Mn does not make sense, the percentage of Mn extracted (25) is relatively high.  So putting all this together, one does in fact come up with 0.08 as the percentage of the RDA (of 10 mg for men) delivered by the extraction of 3 g of ground coffee with 100 mL, and 1.28% of the RDA from the same amount of instant coffee (for which one has to assume that 100% of the Fe in the 3 g of instant coffee are consumed).  Would it not be more informative to express these %RDA numbers as a range? And you should state the assumption about the percentage extraction of elements from instant coffee (or give the actual percentages).

 The authors made corrections to the manuscript.

I cannot follow the calculation of the numbers in Table 4.  I’ve just looked at the Fe numbers.  One “3 g + 100 mL of water” cup of ground coffee delivers 38.1 x 3 x 0.072 µg of iron and so for a 70 kg adult the EDI is 0.12 µg per kg body weight.  The corresponding calculation for instant coffee gives 1.84 µg per kg body weight.  These do not agree with the numbers in the Table.  So some explanation of how the numbers in the Table were calculated is needed.

The authors added information about the % leaching of elements from solutions, which was not visible in the earlier version. We hope that this version is readable

It might be a good idea to check the numbers in Table 5.

The authors made corrections to the manuscript.

Chemical Analysis:  While it is of interest, no doubt, to know the concentrations of these elements in ground and instant coffees, in terms of the impact on the diet, it is the concentrations in the brewed coffee that is important.  First, though, a couple of comments about the analysis of the solids: did you account for the moisture content?  So the results are on a dry weight basis?  What about the reference material (RM)?  The certificate values will refer to dry weight.  This RM (tea leaves) is not a good match for theses coffees (and the iron value is not a certified value). Leaving aside the difference in matrix, the concentrations in the RM are many times those in your samples (e.g. the Mn content is 1570 mg/kg) and the analysis must have involved considerable dilution.  Some details should be provided.  Did you take 500 mg of the RM?

The authors failed to obtain material derived from coffee. the authors used the INCT-TL material because it is available in the department and has been used for years and is therefore well known. 0.2 g of the sample was weighed. Additionally, several other materials available in the unit were tested before performing the actual analyses. In the INCT-TL material, the value for iron has only informative value, therefore samples fortified with a standard iron solution were added. 

More importantly, details of the analysis of the extracts should be provided.  Were they filtered? Diluted? And what were the concentrations of elements found (in each of the 8 samples?).  How do you know these results are accurate and that there were no matrix effects?  The results of spike-recovery experiments should be provided.  Results should be presented to show that 100% of the elements are, in fact, extracted from instant coffee.

The contents were filtered if necessary (using a Whatman No. 40 filter). The samples were diluted to a volume of approximately 25 ml due to the need to dilute the acids used. When calculating the final results, dilutions were taken into account.

As a postscript, estimating the daily dietary intake based on one “cup” made from 3 g of coffee is surely a significant underestimation of the the reality.  Many individuals will surely consume the extract from a much higher mass?  As an example, I drink relatively weak coffee, but still consume the extract of about 15 g of ground coffee (a mixture of caffeinated and decaffeinated) per day (in a total volume of about 900 mL).

The authors have made appropriate corrections. Following the reviewer's suggestion, it was decided to also convert coffee infusions to the amount of 6.33 g given in the statistical yearbook for Poland [15]

And finally, what about decaffeinated coffees?  Does the extraction of the caffeine impact the mineral content?

When selecting the types of coffee for analysis, the authors took into account consumer preferences, and Polish people mainly drink coffee made from ground coffee beans and instant coffee. . Of course, it is worth exploring other types, but this will be the subject of further analysis.

Comments on the Quality of English Language

For the most part, the English is fine.  The confusion between "amount" and "concentration" needs to be fixed and there are some errors.  For example, on line 251, the word "copper" should be manganese.

Thank you for pointing this out, the authors have corrected the manuscript.

Reviewer 3 Report

Comments and Suggestions for Authors

This short and straightforward manuscript describes the trace element content of a variety of types of commercial coffee from Poland.  The manuscript is generally well written with the methods used carefully described. The work appears to be competently performed.  

The manuscript has minor grammatical errors scattered throughout.

Repetition in lines 287 - 292 should be eliminated.  Lines 34 - 37 could also be reworded to eliminate repetition and apparent discrepancies.

In line 87, 0.5 +/- 0.001 might be better written as 0.500 +/- 0.001. 

Comments on the Quality of English Language

The manuscript has minor grammatical errors scattered throughout, but these could be corrected during the proof process.

Author Response

Dear Reviewer,

all authors would like to thank all the Reviewers and Editor for their comments and time spent on the manuscript evaluation. Thank you for giving us the opportunity to revise and improve our article. All the changes have been saved in blue in main manuscript. We hope that our answer will be sufficient to dispel the reviewer's doubts.

The text has been improved and checked by native speaker.

  1. Repetition in lines 287 - 292 should be eliminated.  Lines 34 - 37 could also be reworded to eliminate repetition and apparent discrepancies.

Corrected as suggested

  1. In line 87, 0.5 +/- 0.001 might be better written as 0.500 +/- 0.001. 

Corrected as suggested

  1. Comments on the Quality of English Language

The manuscript has minor grammatical errors scattered throughout, but these could be corrected during the proof process.

The manuscript was proofread by a native speaker

Reviewer 4 Report

Comments and Suggestions for Authors

Manuscript ID: foods-3082733

Title:  Tracking trace elements found in coffee and infusions of commercially available coffee products marketed in Poland

The study investigated coffee as a source of nutritionally important and toxic elements. Specifically, the study explored the coverage of the recommended daily allowance of iron, zinc, copper, and manganese by a cup of ground or instant coffee. Additional goal of the study was the evaluation of the safety of the coffee brew in terms of the intake of toxic elements, lead and cadmium.

The work fits the journal scope. However, the manuscript is not written in a well-structured manner and there are many repetitions of the results. The methods are not described with sufficient details. The data used for calculations are not accurate. The statistical analyses are appropriate but statistical reporting need improvement. The results are interpreted consistently throughout the manuscript, but repeated many times. According to the Data Availability Statement, raw data are available on request. The tables should be more informative and based on the accurate data. Cited references are mostly recent publications and relevant, without significant number of self-citations. The study conclusions need to be revised.

Comments:

Lines 21-22:  A cup of coffee provides x-y% of RDA value - depending on the microelement.

Lines 23-25: There is no PTWI for lead, and the one used for Cd should be updated (see further comments)

Lines 110-111: the matrix of the certified reference material must be identified in order to support the fitness of the quality control procedure.

Lines 118-121: … the infusions were prepared according to the manufacturer's instructions, and the results compared with…. – the authors need to provide information related to the determination of elements in the prepared infusions and calculation of the transfer rate –results in lines 183-184.

Lines 121 and 186: the authors refer to the Polish standard (22) in the Materials and methods section, as well as in the Result section, however, these values are given only in Discussion section of the manuscript. Considering that these values are necessary for the calculations of the results which are the main aim of this study, the authors need to introduce these values much earlier in the manuscript.

Lines 130-131: C – content of the element in the analyzed coffee (mg/kg) – in coffee or in prepared coffee drink? According to Table 3, it is a cup of coffee, meaning drink.

Lines 139-140: there is no established RfD value in EPA IRIS for Pb (please use updated versions of the documents).

Line 149: the formula (4) has to be revised - the authors actually calculated % contribution of the exposure to toxic elements to their respective toxicological doses used for risk assessment.

Lines 149-151: there is no established PTWI value for Pb – it would be appropriate to use margin of exposure approach and BMDL value established by EFSA. For Cd, it would be appropriate to use TWI value established by EFSA.

Line 165, Table 2: the sample collection contains 8 brands, 5 lots each, in total 40 samples. However, Table 2 (and further description of the results) present the ranges of the results for which it is not known are there related to the 4 brands of ground and 4 brands of instant coffee or to the 20 samples of ground and 20 samples of instant coffee… There is no comment about the differences in the results between the lots of one brand. Similar comment for Fig. 1A and 1B.

Lines 167-168 and Fig 1: …. revealed significantly higher levels of iron and zinc in the instant coffee (Fig. 1A) – there are no marks denoting the differences on the Fig. 1A.

Line 192, Table 4: Estimated daily intake is a parameter necessary for calculation of the percentage contribution to the RDA levels of elements and therefore it has to be presented before the %RDA results. The PTWI values for Pb and Cd are already commented – results should be revised.

Lines 195-196: presented numerical values are not connected to the other data presented in the manuscript.

Line 197, Table 5: Table 5 should be replaced so that appears after first mentioning in the text. As there is no RfD value for lead, consequently it is not possible to calculate THQ value for lead.

Line 201: it should be THQs

Line 202: comment …this value is approximately 196 and 32 times lower, respectively (Table 5) … should be related to TTHQ, not to THQ as it is currently implicated.

Lines 205-312: Discussion section should be rewritten – the authors repeat in the discussion all the values presented in the tables of the Results section – there is no need for such duplication of the information.

Line 314, Table 6: this table is part of the results, not discussion. However, it is enough to comment these results in the text, the significance of the information does not justify new table. The issue of the PTWI values for lead and cadmium is already commented.

Lines 325-329: presented information is not result of this study, these are new facts that should not be part of the conclusion.

Line 330: conclusion that coffee is a valuable source … is in contrast with the results of the current study.

Lines 332-336: there is no need to repeat a lot of numbers in the conclusion, but to highlight the most important findings and their importance.

Author Response

Dear Reviewer,

all authors would like to thank all the Reviewers and Editor for their comments and time spent on the manuscript evaluation. Thank you for giving us the opportunity to revise and improve our article. All the changes have been saved in blue in main manuscript. We hope that our answer will be sufficient to dispel the reviewer's doubts.

The text has been improved and checked by native speaker.

Comments:

Lines 21-22:  A cup of coffee provides x-y% of RDA value - depending on the microelement.

Corrected as suggested

Lines 23-25: There is no PTWI for lead, and the one used for Cd should be updated (see further comments)

The authors agree with the reviewer  and replaced manuscript with formulas proposed by the reviewer

Lines 110-111: the matrix of the certified reference material must be identified in order to support the fitness of the quality control procedure.

Information in the manuscript was added

Lines 118-121: … the infusions were prepared according to the manufacturer's instructions, and the results compared with…. – the authors need to provide information related to the determination of elements in the prepared infusions and calculation of the transfer rate –results in lines 183-184.

Information in the manuscript was added

Lines 121 and 186: the authors refer to the Polish standard (22) in the Materials and methods section, as well as in the Result section, however, these values are given only in Discussion section of the manuscript. Considering that these values are necessary for the calculations of the results which are the main aim of this study, the authors need to introduce these values much earlier in the manuscript.

Corrected as suggested

Lines 130-131: C – content of the element in the analyzed coffee (mg/kg) – in coffee or in prepared coffee drink? According to Table 3, it is a cup of coffee, meaning drink.

Information in the manuscript was added

Lines 139-140: there is no established RfD value in EPA IRIS for Pb (please use updated versions of the documents).

The formula has been removed from the manuscript

Line 149: the formula (4) has to be revised - the authors actually calculated % contribution of the exposure to toxic elements to their respective toxicological doses used for risk assessment.

The formula has been removed from the manuscript

Lines 149-151: there is no established PTWI value for Pb – it would be appropriate to use margin of exposure approach and BMDL value established by EFSA. For Cd, it would be appropriate to use TWI value established by EFSA.

The authors apologize for using non-binding standards - they did it because many authors still use them. After discussion, they removed this feature completely and fully followed the reviewer's suggestions.

Line 165, Table 2: the sample collection contains 8 brands, 5 lots each, in total 40 samples. However, Table 2 (and further description of the results) present the ranges of the results for which it is not known are there related to the 4 brands of ground and 4 brands of instant coffee or to the 20 samples of ground and 20 samples of instant coffee… There is no comment about the differences in the results between the lots of one brand. Similar comment for Fig. 1A and 1B.

Corrected as suggested

Lines 167-168 and Fig 1: …. revealed significantly higher levels of iron and zinc in the instant coffee (Fig. 1A) – there are no marks denoting the differences on the Fig. 1A.

To make the results more readable, the figure was converted into a table, and the text was changed

Line 192, Table 4: Estimated daily intake is a parameter necessary for calculation of the percentage contribution to the RDA levels of elements and therefore it has to be presented before the %RDA results. The PTWI values for Pb and Cd are already commented – results should be revised.

Corrected as suggested

Lines 195-196: presented numerical values are not connected to the other data presented in the manuscript.

Corrected as suggested

Line 197, Table 5: Table 5 should be replaced so that appears after first mentioning in the text. As there is no RfD value for lead, consequently it is not possible to calculate THQ value for lead.

Corrected as suggested

Line 201: it should be THQs

Corrected as suggested

Line 202: comment …this value is approximately 196 and 32 times lower, respectively (Table 5) … should be related to TTHQ, not to THQ as it is currently implicated.

Incorrect information removed

Lines 205-312: Discussion section should be rewritten – the authors repeat in the discussion all the values presented in the tables of the Results section – there is no need for such duplication of the information.

Corrected as suggested

Line 314, Table 6: this table is part of the results, not discussion. However, it is enough to comment these results in the text, the significance of the information does not justify new table. The issue of the PTWI values for lead and cadmium is already commented.

Corrected as suggested

Lines 325-329: presented information is not result of this study, these are new facts that should not be part of the conclusion.

Corrected as suggested

Line 330: conclusion that coffee is a valuable source … is in contrast with the results of the current study.

Corrected as suggested

Lines 332-336: there is no need to repeat a lot of numbers in the conclusion, but to highlight the most important findings and their importance.

Corrected as suggested

Round 2

Reviewer 1 Report

Comments and Suggestions for Authors

Although, the authors made some improvements to the manuscript. However, they are not sufficient. The first comments are still valid. Namely, the number of samples - different coffee samples, not repetitions of the same sample, and determined elements is very scarce. Additionally, the aim of the study is superficial. The selection of elements is questionable, why only Cu, Fe, Zn, Mn, Pb, and Cd? There are many reports about the multi-elemental composition of coffee with a much bigger elemental selection and number of samples available. Therefore, the manuscript is still not suitable for publication.

Author Response

Review 1

Although, the authors made some improvements to the manuscript. However, they are not sufficient. The first comments are still valid. Namely, the number of samples - different coffee samples, not repetitions of the same sample, and determined elements is very scarce. Additionally, the aim of the study is superficial. The selection of elements is questionable, why only Cu, Fe, Zn, Mn, Pb, and Cd? There are many reports about the multi-elemental composition of coffee with a much bigger elemental selection and number of samples available. Therefore, the manuscript is still not suitable for publication.

Thank you very much for your insightful review. Fe, Cu, Zn, Mn are among the microelements that perform important functions in the body, as explained in the manuscript. At the same time, their excess is definitely harmful and may lead to the occurrence of various diseases or worsen their symptoms, e.g. manganese deficiency may also limit the absorption of Fe, Zn and Cu. Taking this into account, from the entire spectrum of elements that can be quantitatively determined using selected instrumental methods, the ones mentioned above were selected.

Moreover, in order to determine the coverage of the daily requirement of the Polish population for microelements, reference was made to Polish standards, which contain only elements that perform important functions in the human body. These include microelements selected by the authors. However, the selection of toxic elements was made based on the popularity of these elements in public opinion. We wanted to show that despite their high toxicity, the amounts in which they occur in coffee do not pose a threat to the consumer.

Reviewer 2 Report

Comments and Suggestions for Authors

This revised manuscript is a considerable improvement on the original.  There are still some minor problems, which are probably relatively easily dealt with.

Line  36. Percentage is still a concentration (parts per hundred).  I suggest revising to . . . as well as mineral elements (7%), which may . . . As an aside, these percentages don’t add up to anywhere near 100.  What other compounds does coffee contain?

Line 99. 6.3 ± 0.2 is not what you wish to convey.  I hope that you weighed to the nearest mg (as shown in line 96: 0.500 ± 0.001).  This should read 6.330 ± 0.001 g.

Lines 99 – 101.  It is not clear whether this is ground coffee or instant coffee.

Lines 101 – 106. It is not clear whether this is ground coffee or instant coffee.

I suggest moving the information about the DI water.

For both procedures, the final volume taken to which the solution was made up should be given.

Line 123.  This comment about the final volume also applies here.  A “bottle” is not considered quantitative volumetric ware.  If this was actually a calibrated 50-mL flask, then the text should be amended accordingly.

Line 138 to 139.  My comment about the iron content of the reference material (RM) has been misunderstood.  This material  contains about 432 mg kg-1, but this value is not a certified value—it is an “information” value. 

Lines 136 – 145.  Somewhere in this paragraph should be the details of how the reference material was handled (including information about obtaining results on a dry weight basis).  Maybe even mention of the fact that the concentrations of the relevant elements in the RM are much higher than those in the coffee.  Needs information about the mass taken and the volumes of reagents and the final volume (after any additional dilution).

This comment about whether the results for the solids are presented on a dry weight basis also applies to the analysis of the coffee samples.  It should be clear to readers whether these results are on a dry weight basis or whether they are for samples "as received."

Lines 147 – 148.  It seems unlikely that manufacturers of both ground and instant coffee recommend taking 6.33 g and extracting with 100 mL. Maybe this information is not correct? And so this sentence needs modification?

Line 216.  And also in the discussion.  Copper is an “outlier” in that the concentration in instant coffee is much lower than in the ground coffee.  This should be pointed out at least, if not discussed.

Lines 239 – 241.  The differences in EDI are due to two factors (a) the relative concentrations of each element in each type of coffee and (b) the relative fractions extracted.  You comment is only correct because the concentrations of the elements are about the same (except for that of copper) and a much greater fraction is solubilized from instant coffee.  “Considerably” does not really apply in the case of copper.

Author Response

Review 2

- Line  36. Percentage is still a concentration (parts per hundred).  I suggest revising to . . . as well as mineral elements (7%), which may . . . As an aside, these percentages don’t add up to anywhere near 100.  What other compounds does coffee contain?

 The manuscript was revised according to the reviewer's suggestion

-Line 99. 6.3 ± 0.2 is not what you wish to convey.  I hope that you weighed to the nearest mg (as shown in line 96: 0.500 ± 0.001).  This should read 6.330 ± 0.001 g.

Thank you for pointing this out, of course it was the entry indicated by the reviewer, the manuscript has been corrected

-Lines 99 – 101.  It is not clear whether this is ground coffee or instant coffee.

The manuscript was supplemented with missing information 

-Lines 101 – 106. It is not clear whether this is ground coffee or instant coffee.

The manuscript was supplemented with missing information 

-I suggest moving the information about the DI water.

Information about the water used to prepare the infusions was included in the manuscript on the recommendation of another reviewer, so we apologize but cannot remove it.

For both procedures, the final volume taken to which the solution was made up should be given.

 The manuscript was supplemented with missing information 

Line 123.  This comment about the final volume also applies here.  A “bottle” is not considered quantitative volumetric ware.  If this was actually a calibrated 50-mL flask, then the text should be amended accordingly.

Thank you for pointing this out, the authors did not notice that they had incorrectly entered the container into which the filtered solution was poured - it should have been 25 ml. In addition, they provided information on the transfer of samples after mineralization.

Line 138 to 139.  My comment about the iron content of the reference material (RM) has been misunderstood.  This material  contains about 432 mg kg-1, but this value is not a certified value—it is an “information” value. 

The following information has been added to the manuscript: In order to ensure high quality analyses, in each mineralized series a coffee sample, certified reference material INCT-TL-1 (Institute of Nuclear Chemistry and Technology-Tea Leaves; Polish certified reference material, Poland) (0.150 ± 0.001 g), a blank sample (5 mL 69% HNO3, Merck KGaA, Darmstadt, Germany)and a fortified sample (with the addition of a standard iron solution) were digested. Fortified samples were made due to the lack of Fe in the reference material used. In order to determine the recovery of Fe in ground and instant coffee, an additional 12 samples of instant coffee and 12 samples of ground coffee were prepared (0.500 g±0.001 g), including brands (A-H). 0.1 mL of Fe standard solution (Merck KGaA, Darmstadt, Germany) with a concentration of 0.1 g/L was added to each powder sample (100% recovery corresponds to a concentration of 10 µg/sample). The sample was mixed thoroughly. In order to determine the recovery of Fe in coffee infusions, an additional 12 samples of instant coffee and 12 samples of ground cof-fee were prepared (coffee infusions prepared from 6.330 ± 0.001 g), including brands (A-H) that were fortified by adding 1 mL of Fe standard solution (Merck KGaA, Darmstadt, Germany) to each sample with a concentration of  0.01 g/L (100% recovery corresponds to a concentration of 100µg/100 mL). Samples of certified reference material and fortified samples were mineralized in parallel using the same procedure.

Lines 136 – 145.  Somewhere in this paragraph should be the details of how the reference material was handled (including information about obtaining results on a dry weight basis).  Maybe even mention of the fact that the concentrations of the relevant elements in the RM are much higher than those in the coffee.  Needs information about the mass taken and the volumes of reagents and the final volume (after any additional dilution).

This comment about whether the results for the solids are presented on a dry weight basis also applies to the analysis of the coffee samples.  It should be clear to readers whether these results are on a dry weight basis or whether they are for samples "as received."

The manuscript was supplemented with missing information. 

Lines 147 – 148.  It seems unlikely that manufacturers of both ground and instant coffee recommend taking 6.33 g and extracting with 100 mL. Maybe this information is not correct? And so this sentence needs modification?

Thank you for pointing this out, this information has been corrected

Line 216.  And also in the discussion.  Copper is an “outlier” in that the concentration in instant coffee is much lower than in the ground coffee.  This should be pointed out at least, if not discussed.

 The manuscript was revised according to the reviewer's suggestion.

Lines 239 – 241.  The differences in EDI are due to two factors (a) the relative concentrations of each element in each type of coffee and (b) the relative fractions extracted.  You comment is only correct because the concentrations of the elements are about the same (except for that of copper) and a much greater fraction is solubilized from instant coffee.  “Considerably” does not really apply in the case of copper.

Thank you for pointing this out, this information has been corrected

Reviewer 4 Report

Comments and Suggestions for Authors

Manuscript ID: foods-3082733R1

Title:  Tracking trace elements found in coffee and infusions of commercially available coffee products marketed in Poland

The authors greatly improved both the results and their presentation in the manuscript. There are still several minor issues that need to be resolved.

Line 195: presented PTMI value is for Cd not for Pb – please correct the mistake

Lines 211, 213 and 220: presented values are mean values, so it should be written -mean content- instead of -content-.

Line 234, Table 3: this table should be moved after the current table 4 (the numberings in the tables titles will need corrections, correct also citations of the tables in the text accordingly). Additionally, a short comment of the results presented in table 3 (current numbering) should be given in front of the table, with mentioning the table number in the text.

Lines 242-244: these sentences should be given as separate paragraph after the tables 4 and 3 (current numbering, positioned order as written) and before the text starting in line 248. Note: in the brackets only tables 5 and 6 should be mentioned, not table 4.

Line 253: the authors should add a short comment related to TTHQ results presented in table 5.

Author Response

Review 4 

Line 195: presented PTMI value is for Cd not for Pb – please correct the mistake

Thank you for pointing this out, the manuscript has been corrected

Lines 211, 213 and 220: presented values are mean values, so it should be written -mean content- instead of -content-.

Thank you for pointing this out, the manuscript has been corrected

Line 234, Table 3: this table should be moved after the current table 4 (the numberings in the tables titles will need corrections, correct also citations of the tables in the text accordingly). Additionally, a short comment of the results presented in table 3 (current numbering) should be given in front of the table, with mentioning the table number in the text.

Thank you very much for your suggestion to change the location of table 4, but we will keep it in its current location. Our decision results from the fact that we planned to first show the advantages of coffee consumption (supplementing the diet with essential microelements), and then, in the next point, the danger resulting from the presence of toxic elements in it.

The authors have supplemented the description of table 3 in the manuscript

Lines 242-244: these sentences should be given as separate paragraph after the tables 4 and 3 (current numbering, positioned order as written) and before the text starting in line 248. Note: in the brackets only tables 5 and 6 should be mentioned, not table 4.

Thank you for pointing this out, the manuscript has been corrected

Line 253: the authors should add a short comment related to TTHQ results presented in table 5.

Thank you for pointing this out, the manuscript has been updated.